# Lipid Toxicity in the Cardiovascular-Kidney-Metabolic Syndrome (CKMS)

**DOI:** 10.3390/biomedicines12050978

**Published:** 2024-04-29

**Authors:** John A. D’Elia, Larry A. Weinrauch

**Affiliations:** Kidney and Hypertension Section, E P Joslin Research Laboratory, Joslin Diabetes Center, Department of Medicine, Beth Israel Deaconess Medical Center, Harvard Medical School, Boston, MA 02215, USA

**Keywords:** cardiovascular-kidney-metabolic syndrome (CKMS), lipid toxicity, kidney outcomes, cardiovascular outcomes, PCSK-9 inhibitors, SGLT-2 inhibitors, GLP-1 receptor agonists, small interfering RNA, ceramides, cardiorenal, lipoproteins

## Abstract

Recent studies of Cardiovascular-Kidney-Metabolic Syndrome (CKMS) indicate that elevated concentrations of derivatives of phospholipids (ceramide, sphingosine), oxidized LDL, and lipoproteins (a, b) are toxic to kidney and heart function. Energy production for renal proximal tubule resorption of critical fuels and electrolytes is required for homeostasis. Cardiac energy for ventricular contraction/relaxation is preferentially supplied by long chain fatty acids. Metabolism of long chain fatty acids is accomplished within the cardiomyocyte cytoplasm and mitochondria by means of the glycolytic, tricarboxylic acid, and electron transport cycles. Toxic lipids and excessive lipid concentrations may inhibit cardiac function. Cardiac contraction requires calcium movement from the sarcoplasmic reticulum from a high to a low concentration at relatively low energy cost. Cardiac relaxation involves calcium return to the sarcoplasmic reticulum from a lower to a higher concentration and requires more energy consumption. Diastolic cardiac dysfunction occurs when cardiomyocyte energy conversion is inadequate. Diastolic dysfunction from diminished ATP availability occurs in the presence of inadequate blood pressure, glycemia, or lipid control and may lead to heart failure. Similar disruption of renal proximal tubular resorption of fuels/electrolytes has been found to be associated with phospholipid (sphingolipid) accumulation. Elevated concentrations of tissue oxidized low-density lipoprotein cholesterols are associated with loss of filtration efficiency at the level of the renal glomerular podocyte. Macroscopically excessive deposits of epicardial and intra-nephric adipose are associated with vascular pathology, fibrosis, and inhibition of essential functions in both heart and kidney. Chronic triglyceride accumulation is associated with fibrosis of the liver, cardiac and renal structures. Successful liver, kidney, or cardiac allograft of these vital organs does not eliminate the risk of lipid toxicity. Lipid lowering therapy may assist in protecting vital organ function before and after allograft transplantation.

## 1. Introduction

Large studies have demonstrated that high serum lipid concentrations are risk factors for cardiac and renal disease. Scientific proof of mitigation of atherosclerotic risk by decreasing serum lipid concentrations had to wait for development of relatively inexpensive, effective, and tolerable medications. Studies in large populations support the lowering of certain lipid constituents to decrease morbidity and mortality once atherosclerosis is diagnosed by a sentinel event (secondary prevention) and in certain high-risk populations (primary prevention). Lowering LDL cholesterol levels by 3-hydroxy-3-methylglutaryl coenzyme A (HMG CoA) inhibition with statins or intestinal cholesterol absorption by ezetimibe are established treatments. Other proven therapies include circulating proprotein convertase subtilisin/kexin type 9 (PCSK-9); inhibition by alirocumab, evolocumab, etc.; inhibition by small interfering RNA (inclisiran); or by cholesterol biosynthesis in the liver (bempedoic acid). These medications have been demonstrated to lower LDL cholesterol, with quite significant reductions in mortality and morbidity in high-risk populations. Other medications targeting different pathways (elevating HDL cholesterol, lowering triglycerides, and diminishing apoliproteins) have yet to be demonstrated to have similar benefits.

An increasing prevalence of obesity may blunt the advances that have been made by lowering lipid loads. Therapies directed at morbid obesity include bariatric surgery, glucagon like protein-1 (GLP-1) receptor agonists, or incretin mimetics. Sodium glucose cotransporter-2 (SGLT-2) inhibitors are now beginning to show morbidity and mortality benefits. These agents reduce body weight (visceral fat and muscle) and blood pressure, making it difficult to determine how much survival benefit relates to reduction in serum lipid concentrations alone. The genetic capacity to produce high concentrations of lipids that contribute to chronic ventricular, renal, and hepatic dysfunction, as well as epigenetic factors potentiating obesity, require further study.

## 2. Epicardial Adipose Tissue

Accumulation of excess intravascular triglyceride contributes to adipose deposition within the pericardium and perinephric space. Adipose within pericardium (epicardium) derives blood supply from coronary arteries. Ventricular friction (the summation of contraction, relaxation, torsion, and pulsation of arteries) is cushioned by epicardial adipose tissue [1]. Within the renal capsule, triglycerides are transported in chylomicrons. The minute-by-minute function of the nephron is to filter and reabsorb vital electrolytes, and fuels may be affected by toxic accumulation of lipids [2,3]. Specific pathways have been reported in studies of lipid toxicity to these two vital organ systems, in which injury to one promotes injury to the other. Optimal function of either is required for normal function of the other for control of plasma volume. Volume management in heart failure is a central focus of ventricular and nephron function in CKMS.

Epicardial adipose is active in both lipolysis and lipogenesis. Lipolysis of triglycerides (triacylglycerol) to fatty acids and glycerols is a result of lipoprotein lipase (fasting) and epinephrine or glucagon (post prandial) [4]. The preferred fuel for contraction/relaxation is the fatty acid C-16 unsaturated palmitate. Epicardial adipose serves as the immediately available storage center for beta oxidation down to fatty acids, which are the key component of triacyl glycerol (triglycerides).

Compared with other visceral stores of adipose, epicardial reserves exhibit higher rates of lipogenesis with fatty acid incorporation under the influence of excessive insulin and enhanced activity of local lipase enzymes [2,4]. Myocardial energy utilization is dependent upon certain mitochondrial genes which may be upregulated by peroxisome-activated receptor-gamma coactivator 1 alpha (PPAR gamma) [5]. Although C-16 unsaturated palmitate is the major fuel for cardiomyocyte function, C-16 saturated palmitate is a source of cytokines, and of inflammation, promoting thrombosis and cardiotoxicity. Unsaturated palmitates preserve myocardial cells from apoptosis [1]. Saturated palmitic acid toxicity results from damage to mitochondria, limiting generation of high energy phosphate [6,7,8].

Generation of high energy phosphate in the mitochondria requires transport of fatty acids of various lengths (12 or more carbons) through the double wall of the mitochondrion. (Figure 1a). Figure 1a,b depict the mitochondria as cytoplasmic organelles participating in metabolic pathways (TCA cycle, electron transport chain, and oxidative phosphorylation) used to generate high energy phosphate. Mitochondrial DNA is passed from mother to child.

Long chain fatty acids are transported through the double wall by carnitine palmitoyl CoA transferase. Fatty acids of fewer than 12 carbons do not require carnitine palmitoyl CoA transferase to enter through the inner wall of mitochondria. Oxidation of fatty acids as two carbon fragments (beta oxidation) results in fatty acetyl CoA which enters the tricarboxylic acid cycle for generation of high-energy phosphate (ATP). Figure 1b demonstrates the donation of hydrogens from the reduced form of nicotine adenine dinucleotide (NADH), and the reduced form of flavin adenine dinucleotide (FAD) to the electron transport chain for highly efficient oxidative phosphorylation (Figure 1c) [1]. Peroxisome-activated receptor-gamma coactivator 1 alpha upregulates certain enzymes of the tricarboxylic acid cycle (Figure 1b) [5]. During animal experiments on heart failure, a reduction in mitochondrial function was observed. Progressive loss of efficiency of coupling between the tricarboxylic acid cycle (NADH, FADH) and electron transport (cytochrome C) results in diminished capacity for active ventricular relaxation and contraction [2]. This same loss of efficiency is observed in rodents with experimental diabetes mellitus beginning with normal cardiomyocyte function [11,12]. In an experimental preparation, excess amounts of free fatty acids were associated with mitochondrial apoptosis [13]. Another experimental study found interferon gamma useful in the stimulation of oxidation of fatty acids [14]. Normal epicardial adipocyte function controls movement of electrolytes like potassium and calcium [15]. Excess epicardial adipose or fibrofatty infiltration is associated with increased risk of arrhythmia due to inefficient control of electrolyte movement. Arrhythmias associated with electrolyte imbalance and fibrosis result in common delayed atrial, nodal, or ventricular conduction [15,16]. Excesses of epicardial and abdominal adipose burdens also correlate with elevated risk for coronary atherosclerosis [4]. A derivative of phospholipids (ceramide) is particularly toxic in CKMS [17,18]. In persons with diabetes mellitus, glycation end-products of ceramide are uniquely toxic [19]. Elevation of oxidation/glycation end-products of ceramide sphingolipids have been demonstrated to be associated with the highest risk of adverse cardiovascular endpoints [17], such as heart failure with preserved ejection fraction [18]. Products of redox reactions (oxidized low-density lipoprotein, sphingosine ceramide, and malonyl dialdehyde) are directly toxic in the energy supply of the cardio/renal connection. Balanced energy requirements are critical for continuous calcium flow in cardiomyocytes, as well as for continuous reabsorption of electrolytes/fuels in the renal proximal tubule. Specific injectable treatments to reduce excess LDL are prescribed as alirocumab (Praluent) and evolocumab (Repatha). Myriocin has been used in experimental encephalomyelitis to correct abnormal activity of serine palmitoyl transferase [20]. This observation might carry over to studies on the effect of myriocin on carnitine palmitoyl transferase in the inner wall of mitochondria, oxidizing long chain fatty acids for energy. Aldehydes are highly reactive oxidizing agents. A product of lipid peroxidation, malondialdehyde, was originally useful as a marker of oxidative stress. However, it is so toxic to proteins and DNA that it is considered atherogenic and mutagenic [21]. Individuals with elevated levels of circulating lipids may need pheresis to remove toxic malonyl dialdehyde [22]. This confirms an additional role for pheresis, among centers with expertise in the removal of excessively high levels of lipioproteins for protection of the cardiovascular endothelium [23,24]. A summary of mechanisms of lipid toxicity in CKMS is listed in Table 1.

## 3. Statins, Calcium Channel Blockers, and Early Atherosclerosis

Among hypertensive study subjects with normal cholesterol levels, statins have been shown to be effective in reducing adverse cardiac outcomes [25]. In laboratory studies, calcium channel blockers were synergistic with statins in the prevention of new lesions in carotid and coronary arteries [26]. New lesions in vascular smooth myocytes may involve an excess of calcium intake through L-type calcium channels. Thus, a combination of statin and calcium channel blockers may exhibit synergy in long-term prevention of new atherosclerotic lesions. In different animal and human studies, calcium channel blockers appear to be more effective in the prevention of early atherosclerosis than other anti-hypertensive medications but are not significantly effective once lesions are already progressing [27,28].

## 4. Intra-Nephric Fat

Chronic renal insufficiency is associated with metabolic cardiomyopathy [29,30]. Cholesterol is essential for the structure of the glomerular podocyte slit diaphragm that functions to select molecules for excretion as opposed to retention. In CKMS, excess levels of ceramide sphingolipids can destroy the efficiency of glomerular filtration [19]. A positive correlation has been found between increased concentrations of genes for both triglycerides and apolipoprotein B and the concentration of urine albumin/creatinine [31].

Increased concentrations of both apolipoprotein-1 [32] and oxidized low-density lipoproteins are particularly toxic to blood vessels [33,34,35,36]. The inhibitor of sodium/glucose co-transporter, empagliflozin, has been studied in an experimental mouse model of Alport’s Syndrome. This congenital disorder of the kidney involves accumulation of lipid droplets in the glomerular podocyte. Empagliflozin was shown to reduce the lipid burden in the nephron podocyte [37].

The adverse effect of increased apolipoprotein on the glomerular podocyte in either membranous or diabetic nephropathy has been well studied [38]. Loss of proximal tubular resorption of electrolytes has been associated with excess intra-nephric triglyceride concentration [39]. Triglyceride accumulation is associated with pathologic decoupling of the tricarboxylic acid (Krebs) cycle from electron transfer [13]. Deficiency of PPAR gamma is associated with a genetic marker for inflammation called nuclear factor kappa beta [39]. The leptin-resistant db/db mouse model of type 2 diabetes mellitus, when also deficient in peroxisome proliferator-activated receptor (PPAR gamma) gamma, is associated with development of diabetic glomerulopathy [40]. One mechanism is pathological conversion of epithelium to fibrocytes which can be prevented by thiazolidinediones (pioglitazone, rosiglitazone) [40]. Since leptin-deficiency leads to absence of satiety, the resultant increase in body weight and blood glucose stimulates increased insulin and insulin like growth factor. An important intermediate in the cascade leading to diabetic nephropathy, mammalian target of rapamycin (mTOR), can be inhibited by use of rapamycin [41,42].

Diminished beta-oxidation (two carbon oxidations) of long chain fatty acids may be observed in chronic renal insufficiency. Decreased beta oxidation results in a decreased entry of acetyl Coenzyme A into the TCA cycle which generates high energy phosphate. Thus, there is decreased energy for ventricular relaxation and contraction. The rate-limiting step is the transport of palmitate through the inner mitochondrial membrane under the influence of the enzyme carnitine palmitoyl transferase, which may be imaged using positron emission technology (PET) scans [43].

Chronic renal insufficiency in obese persons with or without diabetes mellitus is associated with disruptions of heart function (cardiometabolic syndrome) [44]. In populations with end-stage renal disease, additional renal toxins not associated with lipids may further impact cardiac function. But not all uremic toxins associated with ventricular hypertrophy/fibrosis are dialyzable. The three best-known non-dialyzable uremia factors associated with heart pathology are indoxyl sulfate, fibroblast growth factor, and beta-2-growth factor. The nuclear gene factor Klotho functions to protect the heart in these situations [29,30]. A summary of mechanisms of lipid toxicity in CKMS is listed in Table 1.

It appears from many multicenter trials that the category of SGLT-2 inhibiting drugs (empagliflozin canagliflozin, dapagliflozin) demonstrates both cardiovascular and renal benefits, although their primary effects were not initially considered antihyperlipidemic. With respect to the cardiac benefits, the outcomes have been measured in events (heart failure, myocardial infarction, stroke, etc.). These effects have been sufficient to incorporate into guideline-directed therapy for heart failure for the ACC/AHA/HFSA and ESC [45,46,47]. The little-cknown effects of empagliflozin, an SGLT-2 inhibitor, include observed increases in ejection fraction and circumferential fiber shortening [48]. Renal benefits have been demonstrated by stabilization of albumin/creatinine and glomerular filtration rate, with prevention of end stage dialysis. With respect to these benefits, both American and European Societies have begun prioritizing the organ protective effects of SGLT-2 inhibition in treatment of type 2 diabetes [49]. Renal benefits have been demonstrated by stabilization of albumin/creatinine and glomerular filtration rate, with prevention of end-stage dialysis. Because SGLT-2 inhibiting drugs reduce serum lipid concentrations through weight loss, and protect glomerular podocytes, we can conclude that elevated lipid levels may be involved in progressive renal dysfunction [19,31,50]. A classical study of cardiac function in hemodialysis subjects involved removing uremic toxins while keeping plasma volume unchanged. This procedure demonstrated improved ventricular function on echocardiography. Left ventricular ejection fraction increased, along with an increased rate of shortening of circumferential fibers. [50]

## 5. Obesity-Related Cardiorenal Pathology in the Setting with Metabolic Liver Disease

Between 2002 and 2016, the observed increase in mean body mass index (BMI) in the United States of America was associated with a rising prevalence of metabolic liver disease [51]. Metabolic liver disease and insulin resistance are inextricably linked to increased hypertensive/cardiovascular/renal risk. Cardiovascular endpoints in populations with metabolic liver disease were 50–75% higher than among matched groups without metabolic liver disease. Likewise, the prevalence of eGFR < 30 mL/min (Grade 5 kidney failure) was 50% higher for the matched population with metabolic liver disease than among matched groups without metabolic liver disease [51,52]. The hazard ratio (HR) for metabolic liver disease with CKD was 1.8; the HR for the relationship between steatohepatitis and CKD was 2.2; and the HR for hepatic fibrosis (cirrhosis) in association with CKD was 3.3.

The most likely intermediate of cardiorenal toxicity is high concentrations of triglycerides in plasma attached to carrier proteins, such as albumin, or in lymphatic fluid as chylomicron particles [25]. The cardiac consequences of the hypertriglyceridemia of metabolic liver disease include myocyte apoptosis and myocardial interstitial fibrosis, with loss of efficiency in contraction/relaxation [53]. When the myocardial metabolic balance shifts from 70% fatty acid oxidation with 30% glucose, amino acid, and lactate as fuel, to a 50/50% distribution, the ventricular contraction and relaxation efficiency is negatively impacted [54]. An alternative to both the slow oxidation of long chain fatty acids (palmitate) and the fast oxidation of glucose would be the intermediate rate oxidation of ketones (acetone, aceto-acetic acid, and beta-hydroxy butyric acid). One of the virtually unpublicized effects of SGLT2 inhibitors is restoration of keto-genesis by insulin, which would have otherwise been inhibited by obesity-related factors such as excess palmitate. Thus, an SGLT-2 inhibitor might be useful in individuals with diastolic heart failure [55].

Another term for metabolic liver disease-related kidney disease would be obesity-related kidney disease [49,56]. The anatomical pathology of obesity-related kidney disease includes generalized enlargement of the glomerulus with both focal and segmental sclerosis. In this setting, damage to peri-tubular capillaries is to be expected [57]. Excess fatty acid accumulation may occur in the setting of high plasma free fatty acids or triglycerides. When the accumulation occurs in the heart, the consequence may be congestive heart failure. When the accumulation occurs in the kidney, the consequence may also be congestive heart failure [54,58,59,60]. High-grade proteinuria (Nephrotic Syndrome) due to increased glomerular basement membrane permeability occurs in association with hyperlipidemia. Specific factors in the pathogenesis of kidney dysfunction include increased oxidized low-density lipoprotein (LDL) and increased intact triglycerides due to a loss of lipoprotein lipase [53]. Oxidized LDL particles are particularly toxic since they stimulate activity of cytokines from macrophages and monocytes [54]. These cytokines initiate inflammation cascades which are nephrotoxic to the chronic interstitial spaces. Chronic interstitial inflammation will lead to fibrosis, an accelerated form of aging [54]. In an obese experimental animal, renal proximal tubule cells developed vacuoles when the diet was switched to high fat. These vacuoles were found to contain phospholipids within structures called lysosomes [61,62]. The best-known toxic phospholipid (sphingomyelin) has a derivative called ceramide sphingosine, which results from loss of the phosphate side chain. A family of potentially toxic phospholipids is outlined in Figure 2. Accelerated kidney aging in (animal or human) subjects with diabetes has been demonstrated through the cross-linking of protein structures by advanced glycosylated end-products. Phospholipids (sphingolipids) linked by advanced glycosylated end-products accelerate the kidney’s aging process to an intense degree [17,18].

Guideline-directed therapy for control of hyperlipidemia, hyperglycemia, and hypertension is frequently in place when novel agents are initiated. Statins are generally required to successfully prevent allograft from aging with interstitial fibrosis, due to hypertension, hyperglycemia, and hyperlipidemia [63]. In an experimental model, angiotensin ll has been shown to cause inflammation/fibrosis. This renal toxic process is blocked by use of angiotensin converting enzyme, angiotensin converting enzyme inhibitors, or angiotensin receptor blockers [63,64]. Regardless of the type of allograft (heart, kidney, or liver) post-transplantation immunosuppression is associated with elevated serum lipid concentrations. The key intermediate in this connection is mTOR. mTOR has been associated with toxicity to both the glomerulus and tubule. The most specific inhibitor of mTOR to date is rapamycin. Efforts to reduce hyperlipidemia involve choice of immune suppressive agents, and standard efforts to reduce cardiovascular risk [49].

One simple estimation of waist circumference multiplied by serum triglyceride level, also known as the Lipid Accumulation Product (LAP), estimates abdominal fat accumulation which, in itself, is a marker of insulin resistance. Use of LAP as a continuous variable has been found to be approximately equal to the better-known Homeostasis Model Assessment (HOMA-IR) for predicting insulin resistance [65,66,67]. Among females, there is a relatively strong relationship between LAP and both concentric and eccentric left ventricular hypertrophy [66]. In an elderly population, higher levels of LAP are associated with higher prevalence of left ventricular hypertrophy [67]. Females with type 2 insulin resistant diabetes and polycystic ovary syndrome (PCOS) have a high cardiovascular morbidity and mortality risk [68,69].

The ovary of an experimental animal exposed to C16 fatty acid (palmitic) demonstrates inhibition of oocyte maturation, a model of PCOS. The model also develops a disruption of ATP generation [70]. This disruption does not involve the tricarboxylic acid (Krebs) cycle or the electron transport system. Two steps in the glycolytic cycle (Table 2) are the focal points of disruption of ATP production in this model of PCOS (females with type 2 diabetes have a higher incidence of cardiovascular morbidity / mortality if they also have insulin resistant polycystic ovary syndrome (PCOS)). The ovary of an experimental animal exposed to excess unsaturated palmitic acid (which may occur with PCOS) has been found to demonstrate inhibition of oocyte maturation, as well as disruption of generation of ATP [68,69,70].

## 6. Normal Contraction/Relaxation Cycle Is Disrupted in Congestive Heart Failure

Calcium is required for the interaction of the proteins actin and myosin in the muscle contraction/relaxation cycle. A protein channel controls the entry of calcium into cardiac myocytes. The storage of calcium occurs in the sarcoplasmic reticulum. Release from storage occurs during muscle contraction against a lower concentration. Transport back into the reticulum during relaxation against a higher concentration requires energy. When toxic lipid concentrations burden the ventricle, the gradual loss of energy for dilation–relaxation will occur before the loss of energy for contraction. This mechanism may be operant in congestive heart failure with preserved ejection fraction. Potassium gradients for propagation of the signal for depolarization/repolarization of the individual cardiomyocyte are also critical for this cycle.

The sarcoplasmic reticulum has an entry and an exit door for the movement of calcium. The protein channel through which calcium enters the cardiomyocyte directs calcium to the exit door of sarcoplasmic reticulum, presumably to be sure there is no risk of overload. The exit door, which is called the ryanodine receptor, then engages troponin inside the muscle. Nowhere is the risk of calcium overload more apparent than in release from endo- or sarcoplasmic storage chambers during acute respiratory/cardiac distress syndromes [72,73].

## 7. Mechanisms of Metabolic Acidosis in CKMS with Diabetes Mellitus

The heart muscle cell (cardiomyocyte) prefers fatty acids for generation of high energy phosphate. The preferred pathway is the tricarboxylic acid cycle which generates reduced nicotine and flavin adenine. These reduced forms provide hydrogens for electron movement in the electron transfer chain. If there is cardiac muscle stress, the slower use of fatty acids in two complex pathways can be exchanged for rapid metabolism of glucose in the simpler glycolytic cycle. Metabolic acidosis can occur from accumulation of acetoacetate/beta hydroxybutyrate when glycolysis is impeded by lack of adequate insulin. Other causes of metabolic acidosis include the accumulation of lactate in hypoxic or septic states and the accumulation of byproducts of uremia associated with renal compromise. In each instance, a metabolic acid requires a buffer in the form of bicarbonate. The loss of an available bicarbonate ion to buffering may be quantitated by direct measure, or by calculating the anion gap. The ketones in diabetic ketoacidosis are known as acetoacetate, beta-hydroxy-butyrate, and acetone. In uremic acidosis, there are dialyzable acids (sulfuric, hippuric, and lactic), partially dialyzable acids (phosphoric), and non-dialyzable acids (indoxyl sulfuric). Acidosis is associated with a prothrombotic state, toxic to endothelial, renal, and cardiovascular integrity [74,75].

Coronary endothelial cells preferentially depend upon glucose for metabolic energy, using the glycolytic cycle with oxidation of fatty acids as a secondary choice in times of pressure/volume, metabolic, and/or ischemic overload. Recent evidence has shown that localized human coronary endothelial atherosclerosis could be influenced by interferon gamma (IFN-Gamma) through t cell lymphocytes.

Interferon gamma is an integral part of the inflammatory process that impairs endothelial glucose metabolism, resulting in a metabolic shift toward increased fatty acid oxidation. This work suggests a novel mechanistic basis for pathological T lymphocyte-endothelial interactions in atherosclerosis [14]. Intermediates in this pathway that inhibit ATP generation include tryptophan and nicotinamide adenine dinucleotide [14]. Mutations in mitochondrial DNA may also cause MELAS (Mitochondrial encephalomyopathy, lactic acidosis and stroke-like episodes) in adult patients [76].

## 8. Continuing Studies

Insulin resistance with dyslipidemia often occurs before elevations of glycohemoglobin A1c. Treatment with the SGLT-2 inhibitor dapagliflozin has been observed to reduce the incidence of new-onset diabetes mellitus in participants with chronic kidney disease or heart failure that involved no reduction in glycohemoglobin A1c [77]. Therefore, interventions in the synthesis of circulating lipids would be expected to reduce the risk of vascular complications associated with uncontrolled glycemia. SGLT-2 inhibitors have been associated with both reduced hepatic synthesis of triglycerides and increased metabolism of lipoproteins. Since lipoprotein (a) has been identified in macrovascular pathology above and below the renal arteries, this represents a future target for SGLT-2 class of medications [33,34]. SGLT-2 inhibitors are not primarily thought of as lipid regulators, because they have only modest effects on serum lipid concentrations, and reductions in morbidity/ mortality are associated with weight loss and glycemia control. Glucagon-like peptide 1 receptor agonists (GLP-1RA) have modest effects on circulating lipids and are usually thought of as beneficial in the treatment of persons with type 2 diabetes, by virtue of glycemia control and weight reduction. When compared to placebo, GLP-1RA (liraglutide or semaglutide) demonstrated stabilization of urine albumin/creatine with a significantly lower loss rate of glomerular filtration [78]. In another study involving patients with eGFR of <30mL/min, a comparison was made between GLP-1RA medications and dipeptidyl peptidase 4 (DPP-4) inhibitors. The results demonstrated significantly lower rates of loss of kidney function or requirement for dialysis for the GLP-1RA study subgroup [33,34,76,77,78,79]. Although additional studies will be required to assess their direct physiologic roles in preventing lipid toxicity, they are currently used in the treatment of persons with type 2 diabetes and chronic kidney disease [80,81]. Several studies of the GLPRA category of medications have demonstrated sufficient reductions in major adverse cardiovascular outcomes to have achieved a Class I recommendation for use in patients with chronic coronary artery disease [82,83].

Micro RNAs are a class of non-coding RNAs that play important roles in regulating gene expression. Micro RNAs associated with LDL contribute to atherosclerosis via an endothelial inflammatory reaction by cytokines from macrophages [84]. Activation of cytokine secretion occurs through a toll-like receptor activation of RNA sensors [85], developing proteins that effect the metabolism of serum lipids. Focus on the LDL receptor has led to the development of inhibitors of proprotein convertase subtilisin/kexin type 9 (PCSK-9) class with further work on small interfering RNA particles demonstrating safety and tolerability [86,87,88]. Medications which reduce synthesis of low-density lipoproteins (alirocumab, evolocumab, etc.) preserve contraction relaxation of the cardiomyocyte through inhibition of interstitial inflammation-fibrosis [53,54,57]. Extracorporeal pheresis may be required in some conditions, for which usual drugs are contraindicated and novel drugs are unavailable.

Although there is strong published evidence in well-designed studies on the efficacy of PCSK-9 inhibition upon reduction in cardiovascular events, there is a relative paucity of studies demonstrating preservation of kidney function. An important contribution to this field has been reported with renal proximal tubular cells, streptozotocin diabetic experimental animals, and with type 2 diabetic study subjects. Investigators found significant preservation of renal function among study subjects with type 2 diabetes mellitus whose levels of PCSK 9 were lower (first tertile) compared to higher (third tertile) [89]. They also noted that mitochondrial DNA was damaged in the presence of PCSK-9 [90].

Study subjects with type 2 diabetes mellitus demonstrated higher concentrations of PCSK-9 than study subjects without diabetes mellitus. Blood pressure and urine albumin/creatinine were significantly higher in the diabetes study group. Blood levels of triglycerides, blood urea nitrogen, and creatinine were significantly elevated in the diabetes vs. non-diabetes study subjects. Elevated glycohemoglobin A1C and levels of insulin (a measure of insulin resistance) were correlated with concentrations of PCSK-9. A vaccine against PCSK-9 contributed to improved measures of glycemia control [11].

Small interfering RNA particles are associated with inhibition of the LDL cholesterol receptor. Inhibition of the LDL receptor may occur through the effect of a small interfering RNA (inclisiran) [91]. Inclisiran inhibits translation of PCSK-9. This proprotein contributes to lysosomal degradation of LDL receptors in hepatocytes, which releases LDL cholesterol into the circulation. Inhibition of the proprotein decreases LDL cholesterol concentration in serum. Continued progress to identify alterations in lipid metabolism that lead to adverse remodeling of the vascular endothelium, myocardium, and nephron is being made. Some of these observations will identify the effect of genetics and epigenetics on the atherosclerotic process, endothelial dysfunction, and organ dysfunction [91]. Further research to determine whether profiling vascular regenerating cell populations is needed [92,93]. In the meantime, we should be moving from factors of risk to factors of causation of CKMS that can be blocked [94].

## 9. CKMS Pathology Associated with Inflammation

Excessive deposits of epicardial and intra-nephric adipose are associated with vascular pathology, fibrosis, and inhibition of essential functions in both heart and kidney. The inflammatory cascade represents a central pathogenetic process in CKMS. The endpoint of chronic inflammation is endothelial, organ fibrosis, and adverse cardiorenal consequences. Chronic triglyceride accumulation is associated with fibrosis of the liver, cardiac, and renal structures. One of the lesser-known responses to angiotensin inhibition is the blockade of the inflammatory cascade. In many instances, a central role for mTOR is in inflammation associated with hyperlipidemia. The importance of rapamycin as an inhibitor of mTOR should be acknowledged in future clinical studies. Given the central role of inflammation in the pathogenesis of interstitial and endothelial injury/fibrosis, medications that inhibit this process are critical to organ preservation in CKMS [95,96,97,98,99,100,101,102,103].

## 10. Conclusions

Treating serum lipid elevations as mere risk factors for future disease is insufficient. Addressing the biochemical consequences of lipid toxicity and its interaction with the inflammatory cascade and fibrosis will protect vital organ function and decrease progressive solid organ dysfunction.

## Figures and Tables

**Figure 1 biomedicines-12-00978-f001:**
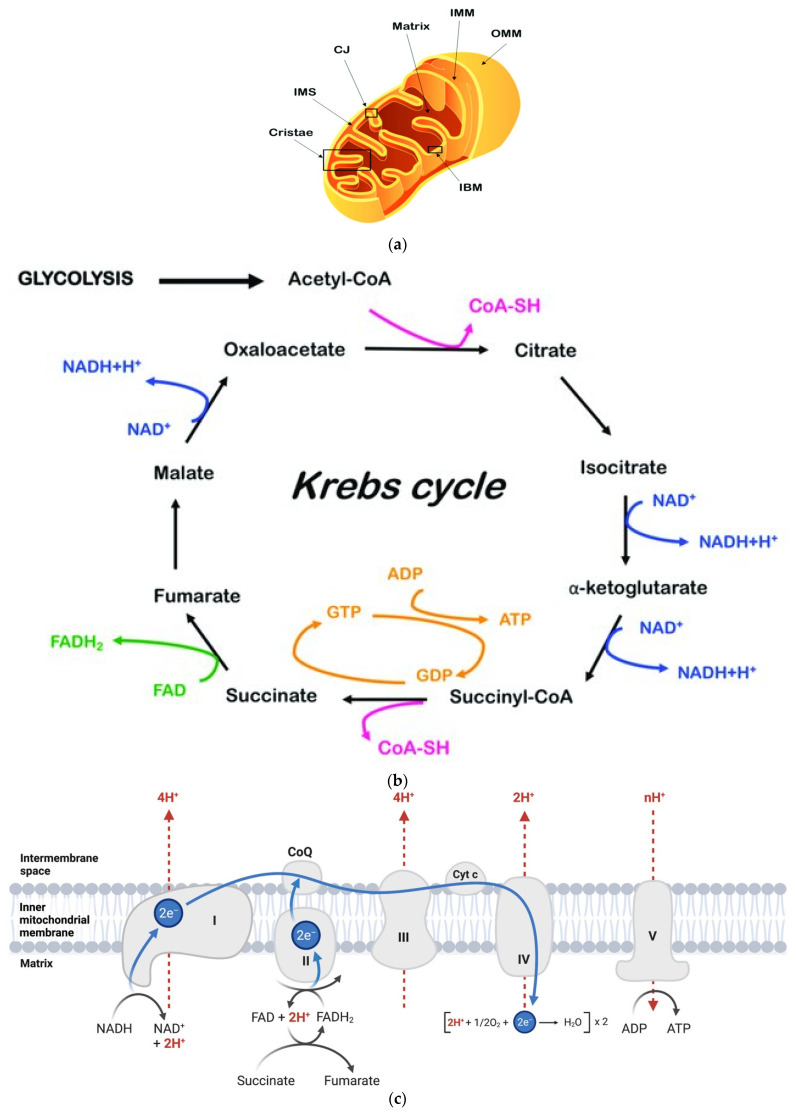
(**a**). **Fatty acids oxidation—mitochondrial structure and bioenergetics in normal and disease conditions:** schematic representation of mitochondrial architecture. The outer mitochondrial membrane (OMM), inner mitochondrial membrane (IMM), inner boundary membrane (IBM), cristae junctions (CJ), intermembrane space (IMS), cristae, and mitochondrial matrix are indicated (From: Protasoni M, Zeviani M. Mitochondrial Structure and Bioenergetics in Normal and Disease Conditions. Int J Mol Sci 2022;22:586) [9]. (**b**). **Schematic representation of the tricarboxylic acid cycle or Krebs cycle.** The Krebs tricarboxylic acid cycle describes a series of reactions by which hydrogen ions (protons) ultimately become available for generation of ATP in the electron transport chain (From: Protasoni M, Zeviani M. Mitochondrial Structure and Bioenergetics in Normal and Disease Conditions. Int J Mol Sci 2022;22:586) [9]. (**c**) **Hydrogen ion transport across the plasma membrane by protein complex releasing ATP energy to cell (oxidative phosphorylation): the mitochondrial electron transport chain** (modified from Yin M, O’Neill LAJ. The role of the electron transport chain in immunity. *FASEB J*. 2021;35:e21974. Doi:10.1096/fj.202101161R Created with BioRender.com) [10]. This illustration indicates electron transfer pathways (blue) and proton translocation (red) in the mitochondrial electron transfer chain. Electrons are fed into the pathway by NADH at complex I (NADH dehydrogenase) and FADH2 from succinate oxidation at complex II (succinate dehydrogenase). Electrons are then transferred to complex III (cytochrome c reductase) by CoQ (a quinone), ultimately through Cyt c to complex IV (cytochrome c oxidase), where it is used to reduce oxygen into water. The transfer of electrons coupled with the translocation of protons across the inner mitochondrial membrane creates an electrochemical proton gradient that drives the synthesis of ATP at complex V (ATP synthase).

**Figure 2 biomedicines-12-00978-f002:**
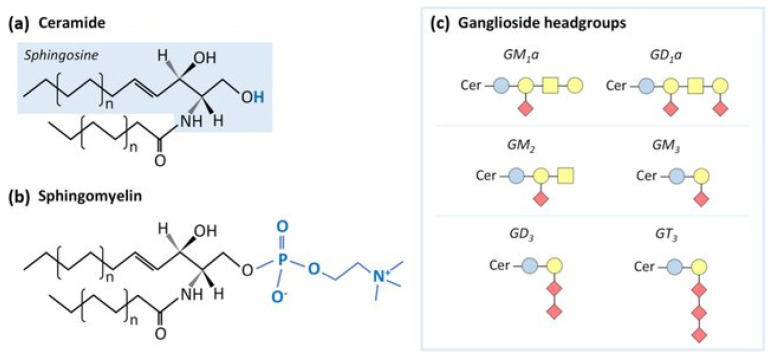
**Simplified structure of phospholipid ceramide sphingosine:** schematic representation of different sphingolipids. (**a**) Ceramide is the building block of more complex sphingolipids. (**b**) Sphingomyelin is one of the most common sphingolipids in mammalian cells. (**c**) Examples of ganglioside headgroups. Gangliosides are glycosphingolipids with a ceramide backbone and headgroups with different sugar unit combinations. Blue circle: glucose; yellow circle: galactose; yellow square: N-acetylgalactosamine; and red diamond: N-acetylneuraminic acid (From: Sarmento MJ, Llorente A, Petan T, Khnykin D, Popa J, Perkovic MN, Konjevod M, Jaganjac M. The expanding organelle lipidomes: current knowledge and challenges Cellular and Molecular Life Sciences 2023; 80(8): DOI: 10.1007/s00018-023-04889-3) [61].

**Table 1 biomedicines-12-00978-t001:** Mechanisms of lipid toxicity in Cardiovascular-Kidney-Metabolic Syndrome.

A. **General:** Lipid Accumulation Product (LAP)	
	combines waist circumference with plasma triglycerides; a marker of abdominal adiposity; associated with cardiac morbidity / mortality.
B. **Heart**	
	Right ventricular contraction / relaxation impeded by epicardial adipose; Cardiac myocyte apoptosis promoted by saturated palmitic acid; Congestive heart failure associated with ceramide sphingolipids; Left ventricular hypertrophy (concentric, eccentric) related to level of LAP.
C. **Kidney**	
	Proximal tubule resorption of fuels and electrolytes impaired by ceramide sphingolipids; Glomerular podocyte efficiency of filtration impaired by the presence of oxidized low-density lipoprotein (LDL).
D. **Both heart and kidney**	
	Hypoxia: diversion of glycolytic acid cycle pyruvate to lactic acid away from the Krebs cycle; Hypotension: diversion of glycolytic acid cycle pyruvate to acetyl CoA for synthesis of fatty acids.

**Table 2 biomedicines-12-00978-t002:** **Summary of glycolytic pathway:** high energy phosphate generated in the last four steps would be the focus of experimental studies of dysfunctional metabolism in the animal model for PCOS, in which generation of ATP is disrupted during a high fat diet phase. Adapted from Bailey, Regina. “Glycolysis”. ThoughtCo, 27 August 2020, https://www.thoughtco.com/steps-of-glycolysis-373394 (accessed on 28 February 2024) [71].

Enzyme	Substrate	Conversion To
Hexokinase	glucose	glucose 6 phosphate
Phosphoglucose isomerase (mutase)	glucose 6 phosphate	Fructose 6 phosphate
Phosphofructokinase	Fructose 6 phosphate	fructose 1,6 bisphosphate
Aldolase	fructose 1,6 bisphosphate	Glyceraldehyde 3 phosphate + dihydroacetone phosphate
Triose phosphate isomerase	dihydroacetone phosphate	glyceraldehyde 3 phosphate
Glyceraldehyde 3 phosphate dehydrogenase	glyceraldehyde 3 phosphate	1,3 bisphosphoglycerate
Phosphoglycerate kinase	1,3 bisphosphoglycerate	3 phosphoglycerate + ATP
Phosphoglycerate mutase	3 phosphoglycerate	2 phosphoglycerate
Enolase	2 phosphoglycerate	phosphoenolpyruvate + H_2_O
Pyruvate kinase	phosphoenolpyruvate	pyruvate + ATP

## Data Availability

Data sharing is not applicable.

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
