# Peer review of "Lipid Toxicity in the Cardiovascular-Kidney-Metabolic Syndrome (CKMS)"

_biomedicines, 2024, doi:10.3390/biomedicines12050978_

Round 1
Reviewer 1 Report
Comments and Suggestions for Authors
I carefully read the review paper titled "LIPID TOXICITY IN THE CARDIOVASCULAR-KIDNEY-METABOLIC SYNDROME (CKMS)". Authors were reviewed the literature to elucidate relationship between lipids and cardiac and renal consequences. They first started emphasizing the toxicity of lipids. The recent studies discussed highlight the detrimental effects of elevated concentrations of certain lipid derivatives on kidney and heart function. Specifically, phospholipids such as ceramide and sphingosine, as well as oxidized LDL and lipoproteins, have been identified as toxic to the vital organs.
Authors also mentioned that energy production was crucial for the resorption of essential fuels and electrolytes in the renal proximal tubule, which is necessary for maintaining homeostasis. In the case of cardiac function, energy derived from long-chain fatty acids plays a key role in ventricular contraction and relaxation. However, excessive concentrations of lipids, including toxic ones, can impede cardiac function. What is the link between anti-oxidant levels and oxidant lipid molecules in term of cardiac function? Discuss please.
Authors also reviewed the metabolism of long-chain fatty acids in cardiomyocytes that occurs through various cycles within the cytoplasm and mitochondria. Toxic lipids and high lipid concentrations can disrupt this process, affecting cardiac contraction. Moreover, proper calcium movement is vital for cardiac function, with contraction and relaxation requiring specific calcium concentrations and energy levels. Please provide more comment on the effects of lipid lowering drugs on cardiac functions.
In another section of the review, authors discussed cardiac functions and its relationship with serum lipids. Diastolic cardiac dysfunction can result from inadequate energy conversion in cardiomyocytes, often exacerbated by factors such as poor blood pressure, glycemic control, or lipid regulation, potentially leading to heart failure. Similarly, disruptions in renal proximal tubular resorption due to lipid accumulation have been observed. Elevated concentrations of oxidized LDL cholesterol are associated with decreased filtration efficiency at the renal glomerular podocyte level. Excessive deposits of adipose tissue around the heart and kidneys are linked to vascular issues, fibrosis, and impaired function in both organs. Chronic triglyceride accumulation can lead to fibrosis in liver, cardiac, and renal tissues. More elaboration of the association between renal functions and lipid levels is needed. Decreased renal function is linked with high levels of serum lipids?
Lipid toxicity is also discussed by the authors which remains as harmful as it is even after transplantation. Therefore, authors highlighted the importance of lipid-lowering therapy which is considered beneficial in protecting organ function both before and after transplantation. Could authors also comment on the possible interaction between lipid lowering therapies and immune suppressive drugs?
All these sections of the paper is well. However, besides the above mentioned revisions, authors should also improve some other parts of the manuscript. First, keywords must be terms or words that mentioned in abstract. So, cardiovascular-kidney-metabolic syndrome (CKMS) does not follow this rule. Second, a section emphasizing inflammation as a pathway in association between lipids, cardiac and renal functions. Serum inflammatory burden is increased in hyperlipidemia (doi: 10.1097/MOL.0000000000000051.), in heart conditions (DOI: 10.1080/17446651.2023.2204941.), and in subjects with deteriorated kidney functions (doi: 10.1159/000368940). Third and finally, drawn conclusions and write a paragraph at the end of the review please.
Author Response
Thank you for your kind and helpful review. We believe that answering your comments has improved the manuscript.
Specifically, we have added discussion regarding the link between anti-oxidant levels and oxidant lipid molecules and cardiac function.
We made further comments on the effect of lipid lowering drugs on cardiac function as well as on the relationship between decreased function and elevated serum lipids
We also commented on the possible interaction between lipid lowering drugs and immune suppressive drugs
We have added a section on inflammation as a pathway
We added a conclusion
We have cited the references that both you and the other reviewer have suggested as well as additional references that were appropriate to the commentary.
We hope that the submission is improved enough to publish
Reviewer 2 Report
Comments and Suggestions for Authors
The present review analyzed the role of lipids and their toxicity in the context of cardiovascular and renal diseases. The authors did an outstanding work analyzing the potential mechanisms involved in the physiopathology of cardiac and renal damage related to lipid accumulation and described new scenarios for the treatment of such conditions. I have no major comments. I'd suggest to add the potential role of lipoprotein aphaeresis among potential treatment for lipid toxicity. Although it is an extracorporeal treatment, it has been and it is still essential in some conditions where there is no chance for novel drugs (PCSK9i for example); in addition, their effects are not only related to lipid lowering approach, but it is also reported in the context of peripheral arterial disease (see PMID: 38398435) in patients with cardiovascular disease to improve clinical outcomes and reduce amputation rate. In addition, there were few cases suggesting the use in nephrotic syndrome to improve proteinuria (see PMID: 30218191) and reduce the extent of renal damage.
Comments on the Quality of English LanguageMinor editing of English language required
Author Response
Thank you or your kind review.
We did add comments regarding the potential role of aphaeresis and included the reference that you have recommended. In addition in response to another reviewer we have added a section on inflammation and a conclusion section. We have also made some changes grammatically to make the paper easier to read (we hope)
Round 2
Reviewer 1 Report
Comments and Suggestions for Authors
Authors are successfully improved the paper as per comments. There are no more issues to suggest further revision. I think the manuscript is acceptable for publication in its current form.